# Parenting of Spanish mothers and fathers playing with their children at home

**Magda Rivero**[1]*, **Rosa Vilaseca**[1], **María-José Cantero**[2], **Esperanza Navarro-Pardo**[2], **Fina Ferrer**[3], **Clara Valls-Vidal**[4], **Rosa M. Bersabé**[5]

1 Department of Cognition, Developmental and Educational Psychology, University of Barcelona, Barcelona, Spain, 2 Department of Developmental and Educational Psychology, University of Valencia, Valencia, Spain, 3 Barcelona City Council, Barcelona, Spain, 4 Department of Psychology, University Abat Oliba-CEU, Barcelona, Spain, 5 Department of Methodology for the Behavioral Sciences, University of Malaga, Malaga, Spain

* mriverog@ub.edu

**Data Availability Statement:** All relevant data are within the paper and its Supporting Information files.

## Abstract

The aims of this study were to compare the parenting behaviors of mothers and fathers when evaluated in a free play situation at home and to study how these behaviors were related to the sociodemographic variables of the family. The study included 155 mothers and 155 fathers from the same families in Spain. The children (90 boys and 65 girls) were typically developing and were aged between 10 and 47 months old. The parents completed a sociodemographic questionnaire, and parenting behaviors in four domains (Affection, Responsiveness, Encouragement, and Teaching) were assessed from self-recorded video-tapes, in accordance with the Spanish version of the PICCOLO. Our results showed both commonalities and differences between the mothers and fathers. The mean scores for the four parenting domains followed a similar pattern in both mothers and fathers: the highest mean score was in the Responsiveness domain, followed by the Affection, Encouragement, and the Teaching domains. Regarding the second aim, no differences were observed in parenting according to the child's gender and the only domain related to the child's age was mother's Teaching. Mothers with a higher educational level scored higher on all parenting domains, except for Responsiveness. Family income was positively related to maternal Affection, Encouragement, and the total PICCOLO score, and to the father's score in the Teaching domain. This study provides evidence that Spanish mothers and fathers show very similar strengths for promoting children's development during interactions. These results are relevant to inform social public policies and family programs.

## Introduction

Parenting is a key topic in research on child development, both from a theoretical point of view and due to its relevance in professional interventions addressed at improving children's developmental processes and family well-being. Although parenting is a wide-ranging concept including many different aspects of childrearing such as parental styles, food and care provision, and parents' beliefs about childrearing, education and development, some authors have

**Funding:** This research was supported by a grant from the Spanish Ministry of Economy and Competitiveness and the European Regional Development Fund (Project PSI2015-63627-R). The funding bodies have not imposed any restrictions on free access to or publication of the research data. All authors are part of the team that received the funding. The funders had no role in study design, data collection and analysis, decision to publish, or preparation of the manuscript. We appreciate the financial aid from the University of Barcelona and the University of Malaga for publishing in open Access.

**Competing interests:** The authors have declared that no competing interest exist.

used terms such as "parenting", "positive parenting" or "developing parenting" to refer specifically to the characteristics of mother/father-child dyadic interactions that promote child development [1, 2]. In this study, we will refer to "parenting" in this sense. Studies on men's parenting are not only increasing but are also going beyond conceptualizations centered on the consequences of paternal absence, men's economic contribution to the family, the father as a source of discipline, and the influence of a male role model. New visions of fathering are focusing on the father's involvement and the role that fathers play in their children's development, behavior, and academic achievement [3, 4]. Most studies on the interactions between fathers/mothers and children have been carried out in the United States, so we found it interesting to be able to provide data from European countries, particularly in those where, as in Spain, the involvement of fathers in the daily lives of children has increased in recent years, but where the model of father as provider and support to the mother as the main caregiver still predominates [5].

With the gradual disappearance of the patriarchal system [6], the incorporation of women into the labor market, the arrival of birth control methods and the legalization of divorce [7] the participation of mothers and fathers in parenting tasks in present-day Spain is far more balanced than in the past. The latest studies show that the involvement of fathers has increased in recent years [8], but this has not led to a decrease in the involvement of mothers. The profile of the involved father is a man with a high level of education, not married and with a partner in full-time employment [9]. Although policies promoting the balancing of family and work commitments in Spain have been relatively weak, in 2021 the duration of paternity leave was raised to the same level as maternity leave in a bid to support equality in parenting. Nonetheless, mothers continue to dedicate more time to childcare [10] and to carrying out more routine parenting tasks, while the most rewarding and socially valued activities tend to correspond to the father [11].

As Cabrera [12] pointed out, mothers and fathers show differences related to the ways they spend time with children. Fathers engage in play activities more frequently than mothers, while mothers tend to engage in more caregiving activities [13, 14]. Some studies have found that fathers tend to engage in more rough-and-tumble play, especially with their sons [13, 15, 16] and that they are more likely to encourage risk taking [17]. A recent review by Valloton [18] about mothers' and fathers' play concluded that mothers and fathers play in a very similar way with little babies, with fathers tending to engage in more highly arousing behaviors. In later infancy and toddlerhood, physical play is more common for fathers and symbolic play is more common for mothers. In the third year, both mothers and fathers (but especially mothers) engage in more symbolic play with daughters, while fathers engage in more physical play with boys.

Particularly at early ages, studies involving children with ages ranging from a few days to 3 years old tend to show no significant differences between the characteristics of the mothers' and fathers' parenting interactions [19, 20]. More recently, a study by Cerezo [21] using state-space dynamic analysis of mothers' and fathers' interactions with their children aged between 6 to 10 months, showed many more similarities than differences, in a sample characterized by low-SES. They analyzed responsive, intrusive, and protective behaviors. The only significant difference was that fathers were more active, having more back-and-forth sequences with their children per unit of time, while mothers showed more repetitions of the same dyadic event. These findings are in line with other previous studies that reported that fathers tend to be more stimulating and activating when interacting with children [22].

However, some studies have found differences between mothers' and fathers' behaviors when interacting with their children [23, 24]. Some researchers have reported that fathers show less responsive behaviors and more intrusiveness than mothers when interacting with their children [25–29]. In contrast, other studies found no differences between mothers and

fathers with respect to responsivity [19, 30]. The above-mentioned review [18] concluded that there is consistent evidence that fathers show less responsive behaviors when interacting with their children at early ages. With respect to intrusiveness, the majority of studies reviewed by Valloton [18] found no differences between mothers and fathers [19]. As Hallers-Haalboom [23] pointed out, studies with comparable methods have reached different conclusions with respect to gender differences in responsivity. So, more research is needed to test the hypothesis that fathers are less sensitive and more intrusive than mothers when interacting with their children at early ages.

When interacting with their children, fathers' talk has been shown to be more directive and informative than mothers, and to include more questions [31]. Some authors have interpreted these characteristics as suggesting that fathers are more goal-oriented than mothers when interacting with their young sons and daughters [23]. Different studies have shown that fathers talk less to children than mothers [24], but not all studies have found this difference [32]. With respect to qualitative characteristics, a study of Italian fathers and mothers reported more lexical diversity and a larger Mean Length of Utterance (MLU) in mothers [33], but the majority of the studies conducted in the United States have shown more similarities than differences in measures of syntactic and lexical complexity [32, 34]. Other studies have mentioned that fathers could be more challenging linguistic partners for young children, by asking more wh-questions and making more requests for clarification [32, 34], by producing more talk that goes beyond the text during book-reading [35] and by using more metatalk in book-reading interactions [36].

With respect to parenting and the child's gender, a meta-analysis [37] found very little evidence to support the notion that parents interact in a different manner with their sons and daughters. A more recent study [38] found that fathers of daughters were more attentively engaged with their child, sang more to them, and used more analytical language and language related to sadness and the body than fathers of sons; in contrast, fathers of sons engaged in more rough-and-tumble play, and used language more focused on achievement. With regards to mothers, some studies have shown that they tend to be more sensitive to their daughters, although when the child presents certain behavioral problems, they have stronger reactions to girls than to boys [39]. Other studies have found that mothers talk more to their daughters than to their sons [31] and that they are more restrictive of physical risk-taking with daughters than with sons [40]. With respect to parental autonomy-supportive and controlling behaviors, a meta-analytic study [41] including boys and girls aged from 0 to 18 years found minimal differences between parental interactions according to the child's gender. Fathers tended to be more restrictive and controlling with their sons than with their daughters, but the effect size was very small, being larger in the youngest group.

In summary, the literature reports both similarities and differences between mothers' and fathers' parental behaviors, with a tendency to find more similarities than differences [42, 43] and also more similarities than differences linked to the child's gender [37, 41]. In general, both mothers and fathers tend to be affectionate, responsive, encouraging and cognitively and linguistically stimulating when interacting with their children [12, 13, 44] and mothers and fathers both make additive and complementary contributions to a child's development. Nevertheless, more research is needed on this topic, in order to increase the diversity of the samples, considering sociodemographic variables and the families' social and cultural contexts.

## Present study

The present study aimed to analyze maternal and paternal parenting during free-play situations at home with typically developing young children in order to compare mothers' and father's behaviors, searching for commonalities and differences. We used the same

observational tool to assess mothers' and fathers' parental behaviors. Although many studies consider this a prerequisite for comparability, this is a condition that not all studies have met [45]. On the other hand, in the present study we focus on free-play situations rather than assessing parenting in a structured task. Free-play situations have been shown to be a context that generates a wider variety of parental behaviors, with parents using more complex language, and allows gender differences to emerge between mothers and fathers when interacting with their sons and daughters at home [18]. Thus, the present study focused on the following aims:

1. To explore how Spanish mothers and fathers of the same typically developing child interact with their son or daughter during free-play situations at home (affective behaviors, responsiveness, autonomy stimulation and non-intrusive behaviors, and cognitive and linguistic stimulation).

2. To analyze the relation between mothers' and father's parental behaviors and some sociodemographic variables related to the child and the parents.

## Materials and methods

### Participants

Participants were 155 mothers and 155 fathers from the same families. All of them were living in Spain. The children ($n$ = 155) were aged between 10 months and 47 months old ($M$ = 28.6, $SD$ = 9.2). Thirty-two per cent of children were younger than 2 years old (10 to 23 months), 43% were 2 years old (24 to 35 months), and 25% were 3 years old (36 to 47 months). Ninety were male (58%) and 65 were female (42%). All children had been born in Spain. Only 3.2% were pre-term and required specialized medical attention at delivery ($n$ = 5). At the time of the evaluation all children were healthy and with normative development as determined by their primary care pediatric history. According to the Bayley Scales of Infant Development (BSID-III) [46] children's percentile scores ranged from 37 to 100 ($M$ = 76.9, $SD$ = 20.5) for cognition, from 18 to 100 ($M$ = 66.5, $SD$ = 27.9) for language, and from 21 to 100 ($M$ = 69.8, $SD$ = 24.6 for motor development). Most of them ($n$ = 121) were attending a nursery or a pre-school center.

Mothers were aged between 22 and 47 years old ($M$ = 35.0, $SD$ = 4.1) and 93.9% were from Spain. One per cent of the mothers had received only elementary schooling, 23% had completed secondary school, 45% had a university degree, and 31% had completed post-graduate studies. Most were in full-time (58%) or part-time employment (32%), whereas 10% were not in paid employment.

Fathers were aged between 25 and 53 years old ($M$ = 37.2, $SD$ = 5.3) and 95.9% were from Spain. Nine per cent of the fathers had received only elementary schooling, 35% had completed secondary school, 35% had a university degree, and 21% had completed post-graduate studies. Most of them were in full-time employment (96%); the rest were employed part-time (3%) or unemployed (1%).

Thirty-two per cent of the families spoke both Catalan and Spanish at home, 30% spoke Catalan, 22% Spanish, and 13% other languages.

Twenty-two per cent of the families had a monthly income between €1,602 and €2,451, considered an average income in Spain [47]. Eight per cent of families had a monthly income below €1,602, and 72% a monthly income above €2,451.

### Instruments

A brief *sociodemographic questionnaire* was used to record sociodemographic variables related to the child (age, gender, attendance at a nursery or a preschool center), and to the parents (gender, age, educational level, working status and monthly family income).

The Spanish version of the Bayley Scales of Infant Development-III (BSID-III) [46] was used to assess the child's development. Cognition, Language, and Motor scales were applied. The Bayley-III has demonstrated high reliability and validity in Spain [46, 48].

The Spanish version [49, 50] of the *Parenting Interactions with Children*: *Checklist of Observations Linked to Outcomes* [1, 2] was used to assess parental behaviors. The PICCOLO is a reliable and validated 29-item measure of parent-child interactions for parents with children aged between 10 and 47 months old. The 29 items reflect parental behaviors linked to the child's developmental outcomes and are scored according to their frequency as 0 (absent, not observed), 1 (rare, minor or emerging) and 2 (clear, definitive, strong and frequent). The items are grouped into four domains: (a) Affection (7 items), which involves the physical and verbal expression of affection, positive emotions, positive evaluation and positive regard; (b) Responsiveness (7 items), which includes reacting sensitively to a child's cues and expressions of needs or interests and reacting positively to the child's behavior; (c) Encouragement (7 items), which considers the parents' support of their child's efforts, exploration, independence, play, choices, creativity, and initiative; and (d) Teaching (8 items), which includes cognitive stimulation, explanations, conversation, joint attention, and shared play. The instrument generates a score for each dimension between 0 to 14 (and 0 to 16 for the teaching dimension) and a total score between 0 and 58. The psychometric properties of the PICCOLO have been found to be satisfactory for both the original and the adapted version [1, 2, 49, 50]. In this study, two trained observers coded 54 mother-child and 29 father-child interactions; inter-rater reliability scores were adequate, and the ICC ranged from .71 to .92. Regarding total scale consistency reliability ($n$ = 155), Cronbach's α value was .84 for mothers and .85 for fathers. With respect to the PICCOLO subscales, Cronbach's α value ranged between .59 and .78 for mothers, and .58 and .73 for fathers.

## Procedure

Ethical approval was obtained from the Bioethics Commission of the first authors' university, according to the guidelines provided by the Council for International Organizations of Medical Sciences (CIOMS), the World Health Organization (WHO), and the World Medical Association (WMA) Declaration of Helsinki—Ethical Principles for Medical Research Involving Human Subjects.

Families were recruited from pediatric centers, nurseries, and Community Family Centers, and they were informed that their participation would be voluntary and anonymous. They received a letter with information about the study, the sociodemographic questionnaire, and a brief guide about how to video record adult-child interactions during play at home. The parents provided signed informed consent prior to participation. Mothers and fathers were asked to record, separately, a play session lasting between 8 and 10 minutes with their child at home in their usual way, with the following instruction: "Interact and play with your child as you normally do". The father and mother could be filmed on video either on the same or different day, within a maximum period of one week. They both chose what to play with their son or daughter. Some games and materials were suggested in the brief guide, for example, books, toy animals, kitchens, dolls, building blocks etc. Mothers and fathers of the same family usually selected different toys when playing with their children, as most of them recorded their videos on different days. Nevertheless, no differences were observed in the type of toys chosen by mothers and fathers, and most adults and children introduced different toys in the play session. Finally, videos were collected and scored according to the PICCOLO criteria by a group of psychologists and specialists in child development. Only videotapes that followed the researcher's instructions (99%) were scored.

## Data analysis

Differences in mean PICCOLO item scores between mothers and fathers of the same child were compared via the Wilcoxon signed-rank test for paired samples. Differences in mean domain and total PICCOLO scores between mothers and fathers of the same child were compared via Student's *t*-test for paired samples, and effect size was calculated by Cohen's *d*. In addition, the relation between mothers' and fathers' parenting scores was analyzed by computing Pearson's correlation coefficients.

For categorical sociodemographic variables, mean parenting scores were compared via Student's *t*-test (for comparing two means) or via robust Brown-Forsythe ANOVA (for more than two means). The relations between parenting scores and demographic variables were examined via Pearson product-moment correlation coefficients (for continuous variables), or via Spearman correlation coefficients (for ordinal variables). Missing data were handled by pairwise deletion. IBM SPSS (version 24.0 for Windows) was used for all statistical analyses.

## Results

### Mothers' and fathers' parenting

Table 1 presents the descriptive statistics (mean and standard deviations) of the PICCOLO item scores for mothers and fathers separately. Only one of the 29 items (item 5 of the affection domain, "Uses positive expressions with child—words such as "honey", "kiddo" or an affectionate nickname") showed a mean score lower than 1 in both mothers and fathers, which indicates that the corresponding behavior was rarely observed in either parent. The four items with the highest mean ($M > 1.85$) were the same for both mothers and fathers: three items from the affection domain ("Speaks in a warm tone of voice", "Is physically close to child", "Is engaged in interacting with child"), and one item from the responsiveness domain ("Pays attention to what child is doing").

The mothers showed a higher mean score than the fathers ($p < .05$) for three items in the affection domain, three in the responsiveness domain, three in the encouragement domain, and five in the teaching domain (see Table 1).

Table 2 presents the descriptive statistics (mean and standard deviations) for the PICCOLO domain and total scores for mothers and fathers. Scores were computed by dividing the total score by the number of items in each domain (mean score). Thus, all mean scores ranged theoretically from 0 to 2. For both parents, all mean scores ranged between 1 and 2. In other words, both mothers and fathers tended to show positive parenting behaviors (affection, responsiveness, encouragement, and teaching) when interacting with their children.

Table 2 also shows that mothers presented higher mean scores in all domains and higher total PICCOLO scores than fathers. Using Cohen's [51] benchmarks for interpreting effect sizes, the effect for the differences between mothers and fathers in mean affection, responsiveness, and encouragement scores can be considered as small ($d \approx .20$), whereas the effect for the differences in the teaching domain and in the total PICCOLO score can be considered as medium ($d \approx .50$).

The mean PICCOLO scores are shown in Fig 1. The mean scores for the four positive parenting domains followed a similar pattern in both mothers and fathers: that is, the order of the mean scores was the same. For both parents, the highest mean score corresponded to the responsiveness domain, followed by the affection and encouragement domains; and the lowest mean score was in the teaching domain.

The relationship between mothers' and fathers' parenting scores was analyzed by computing Pearson's correlation coefficients, which showed statistically significant positive

**Table 1. Descriptives of PICCOLO item scores for mothers and fathers.**

| Domains and Items | Mothers (*n* = 155) | | Fathers (*n* = 155) | | Wilcoxon | |
|---|---|---|---|---|---|---|
| | *M* | *SD* | *M* | *SD* | *Z* | |
| *Affection* | | | | | | |
| 1. Speaks in a warm tone of voice | 1.88 | .34 | 1.89 | .31 | -0.18 | |
| 2. Smiles at child | 1.64 | .54 | 1.55 | .64 | -1.42 | |
| 3. Praises child | 1.64 | .62 | 1.48 | .70 | -2.38 | * |
| 4. Is physically close to child | 1.95 | .20 | 1.88 | .34 | -2.20 | * |
| 5. Uses positive expressions with child | 0.79 | .88 | 0.56 | .79 | -2.76 | ** |
| 6. Is engaged in interacting with child | 1.94 | .24 | 1.87 | .35 | -1.82 | |
| 7. Shows emotional warmth | 1.78 | .47 | 1.74 | .50 | -0.90 | |
| *Responsiveness* | | | | | | |
| 1. Pays attention to what child is doing | 1.90 | .32 | 1.87 | .33 | -0.73 | |
| 2. Changes pace or activity to meet child's interests or needs | 1.72 | .51 | 1.57 | .60 | -2.33 | * |
| 3. Is flexible about child's change of activities or interests | 1.71 | .52 | 1.54 | .61 | -2.61 | ** |
| 4. Follows what child is trying to do | 1.85 | .38 | 1.77 | .45 | -1.66 | |
| 5. Responds to child's emotions | 1.75 | .47 | 1.55 | .60 | -3.61 | ** |
| 6. Looks at child when child talks or makes sounds | 1.79 | .45 | 1.74 | .53 | -1.21 | |
| 7. Replies to child's words or sounds | 1.81 | .47 | 1.77 | .47 | -0.69 | |
| *Encouragement* | | | | | | |
| 1. Waits for child's response after making a suggestion | 1.65 | .55 | 1.60 | .53 | -0.81 | |
| 2. Encourages child to handle toys | 1.81 | .45 | 1.68 | .55 | -2.37 | * |
| 3. Supports child in making choices | 1.59 | .60 | 1.43 | .68 | -2.44 | * |
| 4. Supports child in doing things on his/her own | 1.61 | .54 | 1.57 | .58 | -0.53 | |
| 5. Verbally encourages child's efforts | 1.30 | .75 | 1.23 | .76 | -1.04 | |
| 6. Offers suggestions to help child | 1.49 | .63 | 1.30 | .75 | -2.65 | ** |
| 7. Shows enthusiasm about what child is doing | 1.74 | .50 | 1.66 | .57 | -1.44 | |
| *Teaching* | | | | | | |
| 1. Explains reasons for something to child | 1.04 | .86 | 0.91 | .87 | -1.53 | |
| 2. Suggests activities to extend what child is doing | 1.61 | .61 | 1.35 | .75 | -3.80 | ** |
| 3. Repeats or expands child's words or sounds | 1.74 | .53 | 1.48 | .63 | -4.43 | ** |
| 4. Labels objects or actions for child | 1.83 | .43 | 1.63 | .60 | -3.86 | ** |
| 5. Engages in pretend play with child | 1.22 | .89 | 1.00 | .92 | -2.52 | * |
| 6. Does activities in a sequence of steps | 1.17 | .91 | 1.08 | .89 | -0.95 | |
| 7. Talks to child about characteristics of objects | 1.41 | .74 | 1.19 | .80 | -2.75 | ** |
| 8. Asks child for information | 1.74 | .53 | 1.65 | .60 | -1.74 | |

Notes

* $p < .05$

** $p < .01$.

correlations between parents in the affection domain ($r$ = .273; $p$ = .001), the teaching domain ($r$ = .408; $p$ < .001) and the total PICCOLO score ($r$ = .276; $p$ < .001). However, the correlation coefficient between mothers' and fathers' scores was not statistically significant in the responsiveness domain ($r$ = .089; $p$ = .271) or the encouragement domain ($r$ = .116; $p$ = .149).

## Parenting and sociodemographic variables

The only statistically significant positive correlation between child age and parenting was found for the teaching domain, $r$ = .158; $p$ = .049, in mothers, indicating that mothers' teaching

**Table 2. Differences between mothers and fathers in PICCOLO mean scores (N = 155).**

| PICCOLO score | Mothers | | Fathers | | $t_{(154)}$ | | Cohen's |
|---|---|---|---|---|---|---|---|
| | M | SD | M | SD | | | d |
| Affection | 1.66 | 0.27 | 1.56 | 0.29 | 3.32 | ** | .27 |
| Responsiveness | 1.78 | 0.27 | 1.68 | 0.32 | 3.12 | ** | .25 |
| Encouragement | 1.59 | 0.38 | 1.49 | 0.39 | 2.52 | * | .20 |
| Teaching | 1.47 | 0.36 | 1.28 | 0.44 | 5.07 | ** | .41 |
| Total | 1.62 | 0.24 | 1.50 | 0.27 | 4.83 | ** | .39 |

Notes

* $p < .05$

** $p < .01$.

behaviors were more frequently observed with older children. However, none of the fathers' parenting domains was significantly related to child age. With respect to child gender, Student's *t*-test for independent samples found no statistically significant differences between boys (*n* = 90) and girls (*n* = 65) in PICCOLO mean domain and total scores, for either mothers or fathers. Nor were any statistically significant differences in the mothers' and fathers' parenting scores found between children who attended a nursery or a pre-school center (*n* = 121) and those who did not (*n* = 27).

With respect to mothers' and fathers' age, no statistically significant correlations were found with parenting scores. In relation to parents' educational level, statistically significant Spearman correlation coefficients were found for mothers for the affection ($r_s$ = .197; *p* = .017), encouragement ($r_s$ = .217; *p* = .009), and teaching ($r_s$ = .187; *p* = .024) domain scores and the total PICCOLO scores ($r_s$ = .270; *p* = .001). Thus, mothers with higher educational levels presented more positive parenting behaviors (except for the responsiveness domain) during their interactions with their children. On the other hand, fathers' educational level was not significantly associated with positive parenting interactions and behaviors.

Mean parenting scores were compared across three groups of mothers' working status using robust Brown-Forsythe ANOVA, for more than two independent means. Differences between mean parenting scores for the three groups of mothers were not statistically

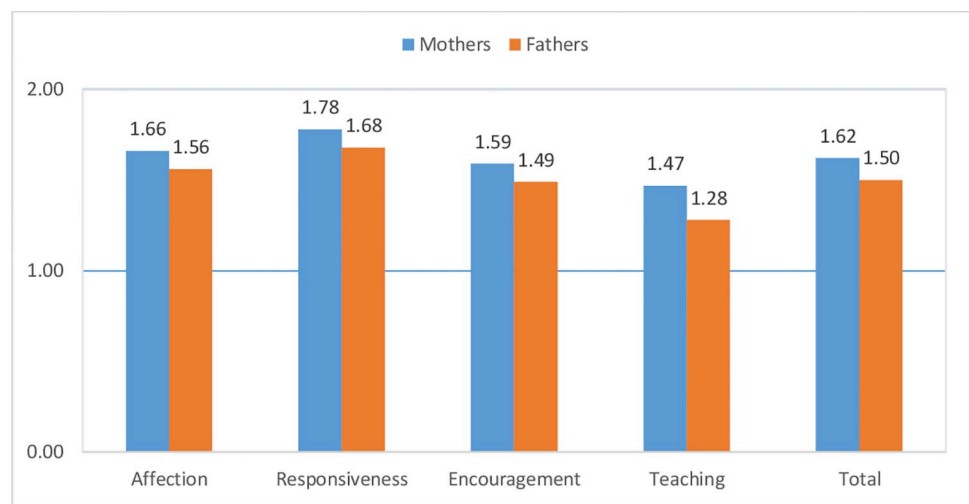

**Fig 1. PICCOLO domain and total mean scores for mothers and fathers.**

significant ($p > .05$). The relationship between fathers' employment and parenting was not analyzed, because the sample size of the part-time employed ($n = 4$) and unemployed/home-makers ($n = 2$) was too small, as most fathers were in full-time employment (96%). With respect to the amount of time that parents dedicate to childcare, the results showed that on workdays mothers dedicated more hours ($M = 6,85$; $SD = 2.93$) than fathers ($M = 4,45$; $SD = 2.94$). The $t$-test for paired samples indicated that the difference between means was statistically significant ($t_{(140)} = 10.31$; $p < .001$) with a large effect size (Cohen's $d = .87$). On the weekend, mothers also spent more hours in childcare activities ($M = 11,62$; $SD = 0.97$) than fathers ($M = 11.25$; $SD = 1.76$), shown by a statistically significant difference ($t_{(140)} = 3.08$; $p = .002$) with a medium effect size (Cohen's $d = .26$). Therefore, mothers spend more time on childcare activities than fathers, although this difference is more pronounced on workdays than on weekends.

With respect to family income, statistically significant Spearman correlation coefficients were found with mothers' affection ($r_s = .184$; $p = .026$), mothers' encouragement ($r_s = .174$; $p = .036$), and mothers' total PICCOLO scores ($r_s = .213$; $p = .010$). In fathers, only teaching domain scores were positively related to family income ($r_s = .234$; $p = .005$).

## Discussion

### Similarities and differences between mothers and fathers in parenting

This study aimed to contribute to the understanding of parenting constructs across gender by including the same measure for mothers and fathers of very young typically developing children. Our first aim was to explore whether mothers and fathers differed in terms of parenting dimensions when they were evaluated in a free-play situation at home. Our results showed both mothers and fathers to be competent in the parental behaviors analyzed, as the mean scores for all parenting dimensions ranged between 1.47 and 1.78 for mothers and 1.28 and 1.68 for fathers, on a scale between 0 and 2. Mothers scored above 1.5 in Affection, Responsiveness and Encouragement and 1.47 in Teaching. Fathers scored above 1.5 in Affection and Responsiveness and 1.49 in Encouragement. Only the Teaching dimension scored below 1.28. These results indicated that the observed parental behaviors were frequent and well established for mothers and fathers during free-play interactions with their children, with Teaching behaviors being somewhat less frequent, especially for fathers. As in previous studies that have compared mothers and fathers by measuring parental positive behaviors with the same tool [52, 53], in our sample mothers scored higher than fathers in a variety of parental behaviors. Mothers scored significantly higher on 14 of the 29 PICCOLO items, in all four PICCOLO domains, and in the total PICCOLO score.

In the Affection domain, mothers praised the child and used positive expressions with him/ her significantly more frequently than fathers and were also more physically close to the child. These results are consistent with those of a review by Abkarian [54], who found that mothers tended to praise their child more, recognizing his/her contributions ("Nice!"). They also corroborate other findings [55] were fathers tend to engage in a more distal style of interaction than mothers, who tend to be closer to the child and to establish body contact with him/her. But these results must be carefully considered, as the size effect of the differences between mothers and fathers in the Affection domain was low.

With respect to Responsiveness, it is relevant that mothers and fathers scored high in all items within the Responsiveness domain, with all the mean scores being above 1.5 out of 2. So, our results are in line with those studies concluding that mothers and fathers are, in general, responsive to the child's signals of attention and interest [19, 30]. In this domain, significant differences were found in three items: "Changes pace or activity to meet child's interests or

needs," "Is flexible about child's change of activities or interests," and "Responds to child's emotions." Our results are in line with those showing that mothers are more responsive to an infant's cues of interest and emotions and that they are more likely to follow the child's lead [27], although the size effect of these differences was low.

With respect to the Encouragement domain, the items showing significant differences were "Encourages child to handle toys", "Supports child in making choices" and "Offers suggestions to help child", and the size effect of the differences in this subscale was also low. Few studies have compared mothers and fathers with respect to positive behaviors encouraging the child's initiative, effort, and autonomy. Some studies [1, 2, 51] using PICCOLO, of parents with a low socioeconomic status in the United States, found that fathers scored lower than mothers in Encouragement, and, in fact, in all PICCOLO domains. It is relevant to mention that this dimension of parenting has been more frequently studied in terms of directive behaviors, control, and intrusiveness. Although behaviors encouraging the child's initiative, effort and autonomy should be considered as the opposite to intrusiveness, it is difficult to compare the results of studies focusing on directiveness and intrusiveness with those analyzing positive behaviors of promotion of the child's autonomy, as in our case. Previous studies about directiveness and intrusiveness found that fathers were more likely than mothers to show intrusive behaviors, such as offering their infant an object while he/she was playing with another one, and to change the way the infant was playing with the object [28]. Kazura [26] found that fathers were more directive than mothers when playing with their children. Our results are somewhat in line with those mentioned, but we must take into account that the size effect of the differences in encouragement was low.

More differences between mothers and fathers were found in the *Teaching* domain, related to the cognitive and linguistic stimulation of children. Mothers scored significantly higher on five items, specifically "Suggests activities to extend what child is doing", "Repeats or expands child's words or sounds", "Labels objects or actions for child", "Engages in pretend play with child" and "Talks to child about characteristics of objects". Both similarities and differences have been identified in previous studies with respect to the language that mothers and fathers use with their child. Among the differences, whereas fathers tend to address the child with more directives and demands for clarification [56] and more open questions [32], mothers tend to repeat or expand the child's utterances and, in general, better attune their speech to the child's language skills. Our data, showing more repetitions and expansions, labels, references to the object's characteristics, and suggestions seem to be consistent with those of Majorano [33] in Italian mothers and fathers. Nevertheless, there were no significant differences in other parental behaviors in relation to the language addressed to the child, such as "Explains reasons for something to child" or "Asks for information". This data would suggest, as other studies have found, that the strengths of fathers when interacting with their children could be those behaviors involving being more challenging for the child, such as questions or talk that goes beyond the context [32, 34–36]. Concerning symbolic play, our results are consistent with those studies reporting that mothers tend to engage more than fathers in pretend play with their young children [18]. We must consider that both mothers and fathers scored lower for teaching behaviors than in the other domains. Some parenting behaviors from the *Teaching* subscale were not very frequent in mother/father-child interactions. These results are consistent with the data obtained when a mothers' sample was analyzed for the validation of the Spanish version of the PICCOLO [49]. Among the different parenting behaviors analyzed using the PICCOLO, teaching behaviors were less frequent than responsive, affective and encouraging behaviors. This was also the case for a sample of Spanish mothers and fathers with a child with intellectual disability [13]. This indicates that the parental behaviors included in this domain should be given special attention in Early Intervention Programs and, in

general, in family interventions oriented to promote positive parenting and the child's development in Spain.

Along with the abovementioned differences, we found some similarities between mothers' and fathers' parental behaviors. As mentioned earlier, the four items showing the highest scores were the same for mothers and fathers. These items were three from the Affection domain ("Speaks in a warm tone of voice", "Is physically close to child", "Is engaged in interacting with child"), and one from Responsiveness ("Pays attention to what child is doing"). This means that these were the most well-established behaviors among mothers and fathers when interacting with their children in free-play situations. Additionally, the order of the PICCOLO subscales, from the highest to the lowest score, was the same for mothers and fathers: Responsiveness, Affection, Encouragement and Teaching. Other studies using PICCOLO reported the same sequence between parenting dimensions with typically developing children [1] and a very similar order (Affection, Responsiveness, Encouragement and Teaching) with children with intellectual disability [13]. In this respect, our results are consistent with those showing more similarities than differences between mothers' and fathers' parental behaviors [44, 57].

With respect to the relationship between mothers' and fathers' parenting scores, significant correlations were found for the affection and the teaching domains, and for the total parenting scores. But there were no significant correlations between mothers and fathers in Responsiveness and Encouragement. Other studies found positive correlations between fathers and mothers in sensitivity and intrusiveness [23, 30]. Some studies indicate that parents can become similar in their parental behaviors as a consequence of cohabitation [58], probably because parents can rely on each other in searching for successful parental strategies [29]. So, our results show both commonalities and differences between the mothers and fathers of our sample. We interpret this as consistent with Cabrera's [12] model of parenting and the transactional models of human development [59], in so far as the parental skills of fathers and mothers would not necessarily be the same for both members of the couple and may to some extent compensate for each other within a family.

## Sociodemographic variables related to parenting

The second aim of this study was to analyze how mothers' and fathers' parenting behaviors were related to certain sociodemographic variables of the child and the parents. When interpreting our results, it must be considered that, in our sample, most of the mothers (76%) and fathers (56%) had completed university studies, and that 72% of the families had above-average incomes. The mothers' and father's educational level and family income have been clearly linked to the quality of parent-child interactions [60, 61]. Halle [62] found that mothers with higher educational levels performed more achievement-related behaviors in their interactions with their children. Parents' educational level has been linked to better verbal engagement with children [63] and to more cognitive stimulation [59]. Family income has been shown to have an impact on the frequency and quality of early literacy experiences at home [64]. Research has also linked mothers' and fathers' education and family income to a warm social climate at home and warm behaviors toward the child [59]. Such data are relevant to inform public policies.

These characteristics of our sample could explain the relatively high scores on the parenting measures, for both mothers and fathers, when compared with the original study of parenting assessed using the PICCOLO in a low-income sample from the United States [1], and with a study in Turkey [65]. Nevertheless, in our study, only maternal educational level (not paternal) was related to parenting. One possible explanation for this lack of relation between the fathers'

educational level and parenting could be that the families that agreed to participate generally had a particular sensitivity towards parenting and education and more egalitarian models of family roles. Additionally, almost all the mothers were employed (58% full-time and 32% part-time), and the fathers even more so (96% full-time and 3% part-time). It is well known that maternal employment is one of the principal causes of social changes in men's and women's roles within the family, including the father's involvement in childrearing [66]. As paternal involvement in childrearing increases, fathers take more responsibility in the tasks of childcare and education [67] and develop a better repertoire of positive parenting behaviors with their children [68], and the role of fathers becomes less stereotypical [33].

It is important to note that the mother's working status (full-time versus part-time) did not affect the analyzed positive parental behaviors. These results are in line with those studies concluding that a mother's number of working hours does not negatively affect some aspects of parenting such as the amount of time spent with her child or her knowledge about her child's characteristics [69]. Nursery attendance was not a factor affecting parental behaviors.

In our study, the age of the child was not related to parental scores in the assessment, with the exception of the score for the mother's Teaching domain. This indicates that some parental behaviors included in the Teaching scale are more frequent when interacting with older children (2 and 3 years old). This is relevant for parental programs.

With respect to the parent's age, research suggests that the emotional stability of older parents is linked to more involvement in parenting and better coping with the stresses linked to parenthood [52]. However, in our sample, there were no differences in parental behaviors according to the parents' age, maybe due to the fact that in Spain, the mean maternal and paternal age is relatively high, at above 30 years [47]. The mean age of the mothers was 35 years and the mean age of the fathers was 37 years in our sample. With respect to the child's gender, our results are in line with those not reporting significant differences in parental interactions [37, 41].

## Conclusions

The main conclusion of our study is that both Spanish mothers and fathers are competent at interacting with their young children in affectionate, responsive, encouraging, and didactic and stimulating ways, performing the positive parenting described in the PICCOLO [2] relating to Affection, Responsiveness, Encouragement and Teaching. Nevertheless, the mothers scored higher than the fathers in all parenting dimensions. This could be explained, at least in part, by the fact that, in Spain, mothers still spend more time than fathers in childcare activities, especially on workdays, as was the case in our sample. In most Western countries, mothers are still the primary caregiver, spend more time on parenting and take more responsibility for family tasks and childrearing than fathers [45].

Our data suggest that mothers and fathers show both similarities and differences in their parental interactions with their children, and that they can compensate for this in each particular family [12]. Our results are in line with the well-established conclusion in the literature that both mothers and fathers can be good parents [70], and show that, in our sample, Spanish mothers and fathers are competent at interacting with their children in ways that promote positive development.

Consistent with previous studies [59, 60], our results show that the mothers' educational level and family income are associated with the quality of parent-child interactions. However, in our study, the fathers' educational level was not associated with parenting ability. This may be due to the non-probabilistic nature of the sampling and the possible predominance of fathers who were particularly strongly involved in the upbringing of their children and highly

aware of the importance of such involvement. Beyond such limitations, we consider that our study contributes valuable data and complements other studies in the field conducted in samples with different characteristics and from different countries.

A practical derivative of our study is that it supports the desirability of incorporating male parents into family intervention programs [13], both in those aimed at families with children with delayed or developmental disorders and the general population. In our study, Spanish fathers were found to perform positive parental behaviors during free-play interactions. These results may be of interest to inform social and family public policies.

## Limitations and future directions for research

This study has several limitations that should be taken into consideration. The first is the selection of the sample. More studies are needed in order to include a wider sample of the Spanish population, showing more variability with respect to educational level, family income, and some other relevant variables such as parents' ideas about co-parenting and gender roles. Such studies will be necessary to inform public policies and family intervention programs. Second, this is a descriptive and transversal study and there were no observational measures of parenting over time. Mothers' and fathers' parenting behaviors may change as times passes depending on factors such as the age of the child, family structure, employment situation of the parents and other socio-demographic factors. Third, we stress that in this study we did not analyze whether the parental behaviors of mothers and fathers predict children's subsequent cognitive, linguistic, or socioemotional development. Although children's development was assessed in this study, both at the time when parental behaviors were recorded and ten months later, the relationship between parenting and child development was not specifically explored. Future studies should continue to examine the effect of mothers' and fathers' parenting on child development applying longitudinal designs, in line with some previous research [71, 72]. We are currently analyzing the relationships between parental behaviors and child development in the sample of the present study. Studying how mothers and fathers can contribute to child development is a very relevant topic, which justifies the interest in increasing the number of studies about parenting of mothers and fathers in European countries. Finally, we have recently begun to transcribe caregiver-child interactions. This new focus opens up a whole new area for research analyzing the quantity and quality of Child Directed Speech. The aim is to identify the aspects of input that contribute the most to children's language acquisition during early development, examining the child's role in interaction, and conducting dyadic analyses.

## Supporting information

**S1 Data.**
(SAV)

## Acknowledgments

The authors would like to thank all participants, parents, and children, and collaborating staff who took part in the research.

## Author Contributions

**Conceptualization:** Magda Rivero, Rosa Vilaseca.

**Data curation:** Magda Rivero, Rosa Vilaseca, Rosa M. Bersabé.

**Formal analysis:** Rosa M. Bersabé.

**Funding acquisition:** Magda Rivero, Rosa Vilaseca.

**Investigation:** Magda Rivero, Rosa Vilaseca, María-José Cantero, Esperanza Navarro-Pardo, Fina Ferrer, Clara Valls-Vidal.

**Methodology:** Magda Rivero, Rosa Vilaseca.

**Project administration:** Magda Rivero, Rosa Vilaseca.

**Resources:** Magda Rivero, Rosa Vilaseca.

**Supervision:** Magda Rivero, Rosa Vilaseca.

**Validation:** Magda Rivero, Rosa Vilaseca.

**Visualization:** Magda Rivero, Rosa Vilaseca.

**Writing – original draft:** Magda Rivero.

**Writing – review & editing:** Magda Rivero, Rosa Vilaseca, María-José Cantero, Esperanza Navarro-Pardo, Fina Ferrer, Clara Valls-Vidal, Rosa M. Bersabé.

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
