## [Decision Letter · Decision Letter 0]

4 Feb 2022

PONE-D-21-24462Parenting of Spanish mothers and fathers playing with their children at homePLOS ONE

Dear Dr. Rivero,

Thank you for submitting your manuscript to PLOS ONE. After careful consideration, we feel that it has merit but does not fully meet PLOS ONE’s publication criteria as it currently stands. Therefore, we invite you to submit a revised version of the manuscript that addresses the points raised during the review process.

We look forward to receiving your revised manuscript.

Kind regards,

Julie Jeannette Gros-Louis, PhD

Academic Editor

PLOS ONE

https://journals.plos.org/plosone/s/file?id=ba62/PLOSOne_formatting_sample_title_authors_affiliations.pdf"

- http://diposit.ub.edu/dspace/bitstream/2445/171784/1/703803.pdf

- https://journals.plos.org/plosone/article?id=10.1371%2Fjournal.pone.0240320

In your revision ensure you cite all your sources (including your own works), and quote or rephrase any duplicated text outside the methods section. Further consideration is dependent on these concerns being addressed.

3. Please ensure that you include a title page within your main document. We do appreciate that you have a title page document uploaded as a separate file, however, as per our author guidelines (http://journals.plos.org/plosone/s/submission-guidelines#loc-title-page) we do require this to be part of the manuscript file itself and not uploaded separately.

5. Please amend your authorship list in your manuscript file to include names of the authors.

6. Figures should not be in both the manuscript, and as separate files.

Reviewers' comments:

Reviewer's Responses to Questions

**Comments to the Author**

1. Is the manuscript technically sound, and do the data support the conclusions?

Reviewer #1: Yes

2. Has the statistical analysis been performed appropriately and rigorously? 

Reviewer #1: Yes

3. Have the authors made all data underlying the findings in their manuscript fully available?

Reviewer #1: Yes

4. Is the manuscript presented in an intelligible fashion and written in standard English?

Reviewer #1: Yes

5. Review Comments to the Author

Reviewer #1: Thank you for inviting my review of this paper. This is an important study for several reasons. First, it provides much needed data on fatherhood. Second, it uses well established methodologies and measures so that the data can be easily interpreted within the context of the existing literature. Third, it is a substantial naturalistic data set and provides a foundation for future work that will likely yield several important findings. Although the results and conclusions are not highly novel, I think that this manuscript will be of great practical interest to the field and fills a gap in the literature.

Introduction:

The introduction is thorough and well written. I appreciated the comprehensive review of research on fathering. The growing literature on mothers and fathers is very complex and the authors navigate inconsistencies with clarity. I was very interested in the section on mother/father talk, but felt that it was a bit misleading given that language input is not measured as a variable in the current study. I would perhaps recommend shortening this section or clarifying that it is not a focus of the current study.

The authors mention very briefly that fathers’ roles in Spain have been shifting in recent years. It might be helpful to have a bit more evidence / context for fathers’ participation in parenting responsibilities as well as a brief discussion of the cultural context or traditional roles for mothers/fathers in Spanish society in the introduction (though I note that it is addressed in the conclusion section).

Methods/Results:

Were all participants monolingual Spanish-speakers? Were various Spanish dialects used in the home?

Play interactions are often influenced by task and context. What were the range of materials selected by mothers and fathers? Did each family use the same materials when playing with the children or is it possible that mothers selected one toy (e.g., puzzle) and father selected a different toy (e.g., book)?

The child’s role in the dyad is also important. Were any child measures collected (e.g., language development, temperament, attention)? If not, this could be briefly mentioned as a limitation to the present study and as a goal for future work with this dataset.

Were any of the videos transcribed? Evaluating the language interaction between parents and children could be a fruitful direction for future research and could be mentioned in the paper.

6. PLOS authors have the option to publish the peer review history of their article (what does this mean?). If published, this will include your full peer review and any attached files.

Reviewer #1: **Yes: **Amy Pace

---

## [Author Response · Author response to Decision Letter 0]

11 Mar 2022

We greatly appreciate the academic editor considerations and the reviewers’ comments and suggestions, which have helped us improve the original manuscript. Next, we respond to every raised point. We mark the academic editor’s and the reviewers’ comments in bold and our responses in plain letter.

The referred line numbers correspond to the revised manuscript with track changes. New fragments of text and new references are highlighted in green in the revised manuscript.

ANSWERS TO ACADEMIC EDITOR

Please ensure that you include a title page within your main document. You should list all authors and all affiliations as per our author instructions and clearly indicate the corresponding author.

It has been included.

We hope we have followed the instructions to the authors well.

We noticed you have some minor occurrence of overlapping text with the following previous publication(s), which needs to be addressed:

- http://diposit.ub.edu/dspace/bitstream/2445/171784/1/703803.pdf

- https://journals.plos.org/plosone/article?id=10.1371%2Fjournal.pone.0240320

In your revision ensure you cite all your sources (including your own works), and quote or rephrase any duplicated text outside the methods section. Further consideration is dependent on these concerns being addressed.

To avoid minor overlapping with the mentioned paper, we have done some changes:

-We have suppressed two lines in the Introduction section (at the beginning of the paragraph in line 79).

-Lines 80-83 have been rephrased and the reference to the mentioned paper (13) has been included in those lines.

-The same reference has been included in lines 499, 514 and 634.

We note that you have stated that you will provide repository information for your data at acceptance. Should your manuscript be accepted for publication, we will hold it until you provide the relevant accession numbers or DOIs necessary to access your data. If you wish to make changes to your Data Availability statement, please describe these changes in your cover letter and we will update your Data Availability statement to reflect the information you provide.

We provide a data file as supplementary material.

Please amend your authorship list in your manuscript file to include names of the authors.

The authors’ names have been included.

Figures should not be in both the manuscript, and as separate files.

The figure (Figure 1) has been deleted from the manuscript and it has been included as a separate file. We indicate in which part of the paper it should be inserted (line 359).

Note:

-If the paper is accepted, we would need to include this text in de Funding section, in addition to what already is written.

“We appreciate the financial aid from the University of Barcelona and the University of Malaga for publishing in open Access.”

So, the final Funding text is:

This research was supported by a grant from the Spanish Ministry of Economy and Competitiveness and the European Regional Development Fund (Project PSI2015-63627-R). The funding bodies have not imposed any restrictions on free access to or publication of the research data. All authors are part of the team that received the funding. The funders had no role in study design, data collection and analysis, decision to publish, or preparation of the manuscript. We appreciate the financial aid from the University of Barcelona and the University of Malaga for publishing in open Access.

ANSWERS TO REVIEWER

Reviewer #1: Thank you for inviting my review of this paper. This is an important study for several reasons. First, it provides much needed data on fatherhood. Second, it uses well established methodologies and measures so that the data can be easily interpreted within the context of the existing literature. Third, it is a substantial naturalistic data set and provides a foundation for future work that will likely yield several important findings. Although the results and conclusions are not highly novel, I think that this manuscript will be of great practical interest to the field and fills a gap in the literature.

We appreciate the comments of the reviewer.

Reviewer #1: The introduction is thorough and well written. I appreciated the comprehensive review of research on fathering. The growing literature on mothers and fathers is very complex and the authors navigate inconsistencies with clarity. 

We appreciate the comments of the reviewer.

I was very interested in the section on mother/father talk, but felt that it was a bit misleading given that language input is not measured as a variable in the current study. I would perhaps recommend shortening this section or clarifying that it is not a focus of the current study.

We agree with the reviewer that this section was too long in the original manuscript and it addressed issues that are not the focus of our study. As suggested, the section about mother/father talk has been shortened. Nevertheless, a part of it has been maintained, given that, although the analysis of language input is not the focus of our study, speech is an important element of adult-child interaction, and the tool that we have used in our study includes some items related to the adult’s talk at the Teaching scale (e.g., repeats or expands child’s words or sounds; labels objects or actions to child; ask child for information). In fact, some ideas and studies referred at this section are considered in the Discussion section, in relation to the Teaching dimension.

Reviewer #1: The authors mention very briefly that fathers’ roles in Spain have been shifting in recent years. It might be helpful to have a bit more evidence / context for fathers’ participation in parenting responsibilities as well as a brief discussion of the cultural context or traditional roles for mothers/fathers in Spanish society in the introduction (though I note that it is addressed in the conclusion section).

In response to the reviewer’s suggestion, we have added a paragraph in the Introduction section about the cultural context of mothers and fathers’ involvement in childrearing in Spain, including some data and some references (lines 67 to 78).

Reviewer #1: Were all participants monolingual Spanish-speakers? Were various Spanish dialects used in the home?

Information about languages has been included in the Participants section (lines 241-242).

Reviewer #1: Play interactions are often influenced by task and context. What were the range of materials selected by mothers and fathers? Did each family use the same materials when playing with the children or is it possible that mothers selected one toy (e.g., puzzle) and father selected a different toy (e.g., book)?

In response to the questions raised by the reviewer, we have introduced more information regarding play materials in the Procedure section (lines 293-297).

Reviewer #1: The child’s role in the dyad is also important. Were any child measures collected (e.g., language development, temperament, attention)? If not, this could be briefly mentioned as a limitation to the present study and as a goal for future work with this dataset.

We agree with the reviewer that the child’s role is very important when analyzing adult-child interactions. 

Child development was assessed with the Bayley-III Scales at the time when parental behaviors were recorded and ten months later. Descriptive data about child development has been included in the Participants section (lines 226-229). Consequently, some lines about the Bayley-III scales have been included in the Instruments section (lines 251-253). 

A reference to our current line of research considering child development have been introduced in the Limitations and future directions for research section (lines 649-656).

Reviewer #1: Were any of the videos transcribed? Evaluating the language interaction between parents and children could be a fruitful direction for future research and could be mentioned in the paper.

We agree with the reviewer about the interest of our data for analyzing language interaction between parents and children. We have added some lines about our future lines of research in the Limitations and future directions for research section (lines 658-662).

Note:

-We have added a recently published paper about the Spanish validation of the PICCOLO with fathers.

---

## [Editor Report · Decision Letter 1]

28 Mar 2022

Parenting of Spanish mothers and fathers playing with their children at home

PONE-D-21-24462R1

Dear Dr. Rivero,

We’re pleased to inform you that your manuscript has been judged scientifically suitable for publication and will be formally accepted for publication once it meets all outstanding technical requirements.

Kind regards,

Julie Jeannette Gros-Louis, PhD

Academic Editor

PLOS ONE
---

## [Editor Report · Acceptance letter]

9 May 2022

PONE-D-21-24462R1 

Parenting of Spanish mothers and fathers playing with their children at home 

Dear Dr. Rivero:

I'm pleased to inform you that your manuscript has been deemed suitable for publication in PLOS ONE. Congratulations! Your manuscript is now with our production department. 

Kind regards, 

on behalf of

Dr. Julie Jeannette Gros-Louis 

Academic Editor

PLOS ONE